# GD2 Expression in Medulloblastoma and Neuroblastoma for Personalized Immunotherapy: A Matter of Subtype

**DOI:** 10.3390/cancers14246051

**Published:** 2022-12-08

**Authors:** Claudia Paret, Arsenij Ustjanzew, Sara Ersali, Larissa Seidmann, Richard Jennemann, Nicole Ziegler, Khalifa El Malki, Alexandra Russo, Arthur Wingerter, Franziska Ortmüller, Angelina Bornas, Pia Charlotte Wehling, Adina Lepădatu, Malte Ottenhausen, Wilfried Roth, Clemens Sommer, Barbara Fliss, Katrin B. M. Frauenknecht, Roger Sandhoff, Jörg Faber

**Affiliations:** 1Department of Pediatric Hematology/Oncology, Center for Pediatric and Adolescent Medicine, University Medical Center of the Johannes Gutenberg-University Mainz, 55131 Mainz, Germany; 2Helmholtz-Institute for Translational Oncology Mainz (HI-TRON), 55131 Mainz, Germany; 3University Cancer Center (UCT), University Medical Center of the Johannes Gutenberg-University Mainz, 55131 Mainz, Germany; 4Institute of Medical Biostatistics, Epidemiology and Informatics (IMBEI), University Medical Center of the Johannes Gutenberg-University Mainz, 55131 Mainz, Germany; 5Lipid Pathobiochemistry, German Cancer Research Center, 69120 Heidelberg, Germany; 6Institute of Pathology, University Medical Center of the Johannes Gutenberg-University Mainz, 55131 Mainz, Germany; 7Institute of Neuropathology, University Medical Center of the Johannes Gutenberg-University Mainz, 55131 Mainz, Germany; 8Department of Neurosurgery, University Medical Center of the Johannes Gutenberg-University Mainz, 55131 Mainz, Germany; 9Institute of Forensic Medicine, University Medical Center of the Johannes Gutenberg University Mainz, 55131 Mainz, Germany; 10National Center of Pathology (NCP), Laboratoire National de Santé, 3555 Dudelange, Luxembourg; 11Luxembourg Center of Neuropathology (LCNP), Laboratoire National de Santé, 3555 Dudelange, Luxembourg

**Keywords:** neuroblastoma, medulloblastoma, GD2, N-glycolyl GM3, racotumomab, dinutuximab, naxitamab, gangliosides, GD2 quantification

## Abstract

**Simple Summary:**

Molecular analyses discussed in Molecular Tumor Boards are expected to improve cancer treatment by identifying tumor-specific alterations. Here, we quantified the expression of the gangliosides GD2 and N-glycolyl GM3 in neuroblastoma and medulloblastoma, two aggressive pediatric tumors. Our data suggest that subtypes of both entities will benefit from an anti-GD2 directed therapy, and that the ganglioside signature can be helpful in the identification of GD2-positive samples. The integration of lipid analysis may help to identify patients who will benefit from personalized immunotherapy with monoclonal antibodies or CAR-T cells.

**Abstract:**

Neuroblastoma (NBL) and medulloblastoma (MB) are aggressive pediatric cancers which can benefit from therapies targeting gangliosides. Therefore, we compared the ganglioside profile of 9 MB and 14 NBL samples by thin layer chromatography and mass spectrometry. NBL had the highest expression of GD2 (median 0.54 nmol GD2/mg protein), and also expressed complex gangliosides. GD2-low samples expressed GD1a and were more differentiated. MB mainly expressed GD2 (median 0.032 nmol GD2/mg protein) or GM3. Four sonic hedgehog-activated (SHH) as well as one group 4 and one group 3 MBs were GD2-positive. Two group 3 MB samples were GD2-negative but GM3-positive. N-glycolyl neuraminic acid-containing GM3 was neither detected in NBL nor MB by mass spectrometry. Furthermore, a GD2-phenotype predicting two-gene signature (*ST8SIA1* and *B4GALNT1*) was applied to RNA-Seq datasets, including 86 MBs and validated by qRT-PCR. The signature values were decreased in group 3 and wingless-activated (WNT) compared to SHH and group 4 MBs. These results suggest that while NBL is GD2-positive, only some MB patients can benefit from a GD2-directed therapy. The expression of genes involved in the ganglioside synthesis may allow the identification of GD2-positive MBs. Finally, the ganglioside profile may reflect the differentiation status in NBL and could help to define MB subtypes.

## 1. Introduction

Gangliosides, sialic acid-containing glycosphingolipids, play an important role in the etiology of cancer [1]. Several tumors exhibit aberrant ganglioside expression, including the expression of gangliosides not found in normal adult tissues but during fetal development, such as GD2, or incorporated from dietary sources, such as N-glycolyl GM3 [2]. GD2 expression is expected in various types of neuroectodermal cancers including neuroblastoma (NBL), melanoma, and non-small cell lung cancer (NSCLC) [3]. Monoclonal antibodies against GD2 (naxitamab and dinutuximab) are standard care for patients with high-risk NBL [4,5]. An increasing number of clinical studies with monoclonal antibodies or chimeric antigen receptor T cells (CAR-T cells) against GD2 are enrolling patients with different tumor entities. Moreover, racotumomab, a therapeutic vaccine triggering an immune response against N-glycolyl GM3, is approved for the treatment of recurrent or advanced NSCLC [6]. Thus, patients discussed in a molecular tumor board (MTB) could benefit from a ganglioside specific analysis. Molecular alterations discussed in MTBs are mainly mutations, copy number variations (CNVs), and fusions identified by genomic and transcriptomic sequencing. Gangliosides are generally not analyzed or discussed because their detection requires specific methodologies. Immunohistochemistry on formalin-fixed, paraffin-embedded (FFPE) samples has a limited utility in analyzing ganglioside expression because lipids are extracted from the tissue by the solvents used in the process. Moreover, because the enzymatic products of several genes are required to interact in a combinatorial manner to guide the production of gangliosides into different series, the use of genomic and transcriptome data to predict a GD2-positive phenotype remains largely unexplored. Sorokin et al. proposed a two-gene signature for the prediction of GD2-positive samples [7]. They demonstrated that the sum of the ganglioside synthase genes coding for Alpha-N-acetylneuraminide alpha-2,8-sialyltransferase (*ST8SIA1*) and Beta-1,4 N-acetylgalactosaminyltransferase 1 (*B4GALNT1*) is a more efficient predictor compared to single ganglio-series biosynthesis genes or randomly selected pairs of lipid metabolic genes.

NBL is the most common extracranial solid tumor occurring in childhood. Despite the use of very aggressive multimodal therapies, the survival rate in high-risk NBL is only 50% [8]. The prognosis of high-risk NBL patients has improved with treatment with monoclonal antibodies against GD2. Anti-GD2 therapy with monoclonal antibodies is currently only approved for the treatment of high-risk NBL, but has not been effective in other GD2-positive tumor entities. As NBL samples are considered GD2-positive, analysis of GD2 expression is not required before the application of anti-GD2 antibodies. However, comparing the level of GD2 expression in other tumor entities with that in NBL could help to identify those tumors that respond best to anti-GD2 therapies.

Medulloblastoma (MB) is the most frequent malignant brain tumor in children and has been classified into four major molecular subgroups with prognostic and therapeutic potential, namely wingless-activated (WNT), sonic hedgehog-activated (SHH), group 3, and group 4, with each group consisting of additional subtypes [9]. Treatment of MB includes surgical intervention, craniospinal irradiation, and adjuvant chemotherapy, which are associated with high morbidity and fail in around 30% of patients [10]. Genomic analysis has not improved MB outcomes so far. In a recent study with 23 MBs, no clinically relevant molecular targets were identified [11]. GD2 has been discussed as a molecular target in MB [12,13]. ^131^I conjugated GD2 antibodies have been evaluated for the treatment of MB, with some patients achieving objective responses [14,15]. Moreover, a phase I clinical study (NCT04099797) of CAR-T cells targeting GD2 is currently enrolling high-grade brain tumors, including MB. So far, however, it is not clear whether all MB or only some subgroups will benefit from a GD2-directed therapy.

In the present study, we used thin layer chromatography (TLC) and liquid chromatography-coupled tandem mass spectrometry (LC-MS^2^) to compare the ganglioside composition of NBL and MB, including N-glycolyl GM3. Moreover, we applied the two-gene signature described by Sorokin et al., to NBL and MB RNA-Seq dataset and compared the expression patterns of the four MB subtypes with other tumor entities and tissues.

Our results confirm a frequent but heterogeneous expression of GD2 in NBL, while GD2 is a potential target only in some MB subtypes. Furthermore, our data suggests that the ganglioside signature may help in the further characterization of MB subtypes.

## 2. Materials and Methods

### 2.1. Patients and Tissues

Surplus fresh frozen tissues from surgery (14 NBL and 16 MB samples) and from autopsy (normal brain) not needed for histopathological diagnosis were used for the analyses. Please refer to the sections Institutional Review Board Statement and Informed Consent Statement for details. Samples 110, 111, 134 are commercially available RNA samples and were purchased from BioCat (Heidelberg, Germany). Residual lipid extracts from mouse liver used in this study had been collected for a previous study [16].

### 2.2. Materials for Lipid Analysis

All organic solvents were of LC-MS^2^ grade. DEAE sephadex A-25, was obtained from Pharmacia Biotech AB, Uppsala, Sweden. Preparative C18 125°A 55–105 µm, was obtained from Waters GmbH, Eschborn, Germany. Gangliosides GM3, GM2, GD2, GD3, and GD1a were obtained from Matreya, USA. Isotopically labelled D5-GM3 and D5-GM1 were purchased from Avanti Polar Lipids, and isotopically labelled D3-GM2 and D3-GD3 from Cayman Chemicals. GM1a was derived from FIDIA, Italy and Cronassial (bovine brain gangliosides containing mainly GM1a, GD1a, GD1b, and GT1b) from the company Dr. Madaus & Co, Cologne, Germany. HPTLC silica gel 60 F254 glass plates, Merck KGaA (Darmstadt, Germany). Plexigum (Poly(Isobutyl methacrylate)), SigmaAldrich (St. Louis, Missouri, USA). Mouse anti-human GD2 IgG2a antibody, clone 14. G2a, cat. #554272, BD Pharmingen (Franklin Lakes, New Jersey, USA). Alkaline phosphatase-conjugated goat anti-mouse IgG+IgM (H+L), cat. #115-055-068, Jackson ImmunoResearch Laboratories Inc. ((Farmington, CA, USA). SIGMAFASTTM BCIP^®^/NBT (5-bromo-4-chloro-3-indolyl phosphate/Nitroblue tetrazolium) tablets, Sigma Aldrich (St. Louis, Missouri, USA).

### 2.3. Lipid Extraction

Tissues were freeze-dried and powdered. Aliquots of the powdered tissue corresponding up to 20 mg dry weight were extracted three times with solvent mixtures of chloroform:methanol:water (step 1 and 2 with a ratio of 10:10:1 and step 3 with a ratio of 30:60:8) at 37 °C in an ultrasonic bath for 15 min each. Supernatants were collected as “raw extract” and the residual pellet was used to determine total protein with a BCA-based assay.

Lipid raw extracts were desalted with C18-columns. The desalted lipids were subsequently split into neutral and acidic lipids with DEAE ion exchange columns and both fractions were subsequently desalted again with C18-columns. Lipid fractions were dried again with a gentle nitrogen gas stream at 37 °C and dissolved in chloroform:methanol:water (10:10:1) corresponding to 4 mg total sample protein per mL.

### 2.4. TLC and Immune Overlay Analysis

Lipids corresponding to 300 µg total sample protein were loaded onto HPTLC plates with a Linomat IV (Camag, Switzerland) and developed in vertical TLC chambers with a pre-run solvent (chloroform:aceton, 1:1), dried, and the running solvent chloroform:methanol:0.2% aq. CaCl2 (45:45:10). Detection of ganglioside GD2 with anti-GD2 antibody was performed using the immuno-overlay technique [17]. In brief, dried TLC plates were mechanically stabilized by a 2 min bath by a solution of 5% Plexigum 28 in chloroform 1:10 diluted in n-hexane. TLC plates were air dried for 20 min and incubated for 30 min with a blocking solution (1% BSA in PBS). Plates were then incubated with a 1:50-dilution of the anti-GD2 antibody in blocking solution overnight at 4°C, washed 4 times 5 min with PBS containing 0.05% Tween 20. Plates were subsequently incubated with alkaline phosphatase-conjugated anti-mouse immunoglobulin antibody diluted 1:500 in blocking solution for 30 min at room temperature. After repeated washing steps as described before, GD2-positive bands were visualized with substrate for phosphatase (one tablet of SIGMAFASTTM BCIP^®^/NBT dissolved in 10 mL distilled water) up to 6 min. Plates were washed with distilled water, dried, and scanned for documentation. In order to visualize all glycosphingolipids, the TLC-plate was destained in an acetone bath for 5 min to remove color from the immune staining and plexigum. After thorough drying in a desiccator, plates were sprayed with orcinol reagent using a Derivatizer from Camag, Switzerland and developed at 120 °C for 7–10 min. Quantification was performed using the ROI-manager analysis tool of the Fiji-imageJ software. Background intensity of the TLC-plate was subtracted from orcinol-positive bands and intensity was compared to gangliosides standard of known concentration.

### 2.5. LC-MS^2^ Analysis of GD2

An aliquot of the acidic lipid extract corresponding to 600 µg sample protein was mixed with isotopically labelled gangliosides D5-GM3, D3-GM2, D5-GM1a, and D3-GD3 in a total volume of 200 µL. A volume of 10 µL spiked sample was injected onto an Aqcuity I class UPLC (Waters, USA) equipped with a Anionoic Polar Pesticide column (5 µm, 150 mm × 2.1 mm, Waters) and run at 30 °C. A HILIC gradient was applied, starting with 100% solvent A (95% acetonitrile, 3% DMSO, 2% water + 0.03% NH4OH) and increasing solvent B to 90% (96% water, 3% DMSO, 1% Formic acid and 40 mM ammonium formiate) within 20 min. Thereafter, 90% B was kept for 4 min before going back to 100% A to re-equilibrate for another 5 min. Gangliosides were monitored with a triple-quadrupole-like tandem mass spectrometer (Xevo TQ-S, Waters, Milford, MA, USA) in negative electrospray multiple reaction monitoring mode. The transition of single deprotonated molecular ions of GM3, GM2, GM1, and GD3 and of double deprotonated molecular ions of GD2 and GD1 to the sialic acids fragments of N-acetyl neuraminic acid (Neu5Ac, fragment *m*/*z*—290.1, for all gangliosides) and of N-glycolyl neuraminic acid (Neu5Gc, fragment *m*/*z* -306.1, for GM3 and GM2 only) were monitored. GGs with the ceramide anchors d32:1, d34:1, d36:1, d38:1, d40:1, d41:1, d42:2, d42:1, d43:1, and d44:1 were taken into account. GM3, GM2, GM1, and GD3 were quantified in relation to their respective deuterated internal standards. GD2, GD1a, and GD1b were quantified in relation to D3-GD3 taking a response factor for GD3/GD2 and GD3/GD1a into account, which was calculated with known concentrations of corresponding standard gangliosides.

### 2.6. Gene Expression Analysis of Tumor Samples

RNA was isolated from fresh frozen tumor samples using the RNeasy Lipid Tissue Kit (QIAGEN, Hilden, Germany) following the manufacturer’s protocol. Quality of the RNA was determined with a Bioanalyzer Device (Agilent Technologies, Santa Clara, CA, USA) and only samples with adequate RIN values were used for further analysis. Reverse transcription was performed with the PrimeScript^TM^ RT Reagent Kit with gDNA Eraser (TaKaRa BIO INC, Kusatsu, Japan). Quantitative RT-PCR was performed using the PerfeCTa^®^ SYBR^®^ Green Fast Mix^®^ (Quantabio, Beverly, USA) in a LightCycler 480 instrument (Roche, Basel, Switzerland). Raw values were normalized to *hypoxanthine-guanine-phosphoribosyl-transferase* (*HPRT)* and two-gene expression was calculated as sum of *ST8SIA1* and *B4GALNT1* normalized expression values. Data is shown as mean ± SD and statistical analysis was analyzed by two-tailed Student’s *t* test (GraphPad Prism software version 9.0.1). *p*-values < 0.05 were considered significant. Primers for qRT-PCR analyses were as follows: *ST8SIA1* forward 5′AGTGACAGCTAATCCCAGCA3′, reverse 5′TGGCTCTGTTCCTGTCTTCA3′; *B4GALNT1* forward 5′CCTTCAGGCAGCTTCTGGT3′, reverse 5′TGCTGTGTTGGTCTGGTAGC3′; *HPRT* forward 5′TGACACTGGCAAAACAATGCA3′, reverse 5′GGTCCTTTTCACCAGCAAGCT3′.

### 2.7. RNA-Seq Data

MB RNA-Seq data was obtained from Gene Expression Omnibus under GEO accession number GSE203174. The associated sample metadata was taken from the Appendix A of [18]. The dataset consists of a total of 86 RNA-seq samples, which are divided into the following MB subtypes: 23 SHH, 6 WNT, 17 group 3, 35 group 4, and 5 normal cerebellum RNA samples.

Further RNA-Seq data and corresponding clinical annotation were obtained in form of the TCGA TARGET GTEx study dataset from UCSC Xena [UCSC. Xena http://xena.ucsc.edu (2016) accessed on 1 June 2022.] containing in total 19,109 samples (divided in the projects GTEx: 7845, TARGET: 734, and TCGA: 10,530 samples). This study data was previously re-analyzed by the UCSC toil RNA-Seq pipeline and thus eliminating batch effect due to different computational processing [19].

### 2.8. RNA-Seq Data Preparation and Principal Component Analysis

The read counts of the MB dataset were normalized by the median of ratios method using the DESeq2 R package (version 1.34.0) [20]. One pseudocount was summed to the normalized count values. The count matrix was then log10 transformed. Genes of the “Glycosphingolipid biosynthesis—ganglio series” (hsa00604) pathway were obtained from the Kyoto Encyclopedia of Genes and Genomes (KEGG) [21]. A Principal Component Analysis (PCA) was performed on the 15 genes of this pathway and visualized with ggbiplot (https://github.com/vqv/ggbiplot; version 0.55 accessed on 1 January 2022). Additionally, several PCAs were performed on genes of related pathways (hsa00600, hsa00601, and hsa00603), and on six genes directly involved in the synthesis of GD2, namely ST3GAL5, *ST8SIA1*, ST8SIA5, B3GALT4, *B4GALNT1*, and B4GALT6. The results of these additional PCAs can be found in Appendix A.

### 2.9. Differential Gene Expression Analysis

To identify significant gene expression differences between MB groups, a differential gene expression (DGE) analysis was performed on the MB RNA-Seq dataset using DESeq2, fitting the negative binomial generalized linear model for each gene and using the Wald test for significance testing. Benjamini & Hochberg correction was used to obtain *p*-adjusted values. Detailed results of the DGE including log2-Fold-Changes, *p*- and *p*-adjusted values for all genes of interest can be found in Appendix A.

### 2.10. Two-Gene Signature for GD2 Quantification

Recently, Sorokin et al., proposed a two-gene expression signature consisting of the ganglioside synthase genes *ST8SIA1* and *B4GALNT1* that serves as a predictor of GD2-positive phenotype [7]. The signature is calculated by the sum of decimal logarithms of normalized expression levels of both genes. We applied this signature on the log10 normalized count matrix of the MB dataset. Boxplots and scatterplots were drawn using ggplot2 package (version 3.3.6). P-values were added to the boxplots by using the ggsignif package (version 0.6.3) [22].

To compare the two-gene signature of the MB groups with other tumor entities and normal tissues, the MB dataset was combined with public gene expression datasets from The Cancer Genome Atlas (TCGA) [23], The Genotype Tissue Expression Project (GTEx), and Therapeutically Applicable Research To Generate Effective Treatments (TARGET) (https://ocg.cancer.gov/programs/target, accessed on 1 January 2022). These projects were extracted from the TCGA TARGET GTEx study dataset from UCSC Xena The “RSEM expected_count” data (which is available as log2(expected_count+1)) was downloaded, backtransformed to integer counts, and divided by the projects into three datasets. For the explicit comparison of the two-gene signature between MB subtypes and NBL, 162 NBL samples were extracted from the study dataset. The raw counts of the MB dataset were merged with the backtransformed count data of each project dataset/ the extracted NBL samples, and normalized by the median of ratios method using the DESeq2 package. Two-gene signature of each combined dataset was calculated as described above. Heatmaps were drawn using the pheatmap package (version 1.0.12).

## 3. Results

### 3.1. Developpement of a LC-MS^2^ Assay for the Detection of N-glycolyl GM3 and Quantification of GD2

To specifically detect and quantify mono- and disialogangliosides (Figure 1A), the fraction of acidic lipids was analyzed by mixed mode hydrophilic interaction chromatography-coupled tandem mass spectrometry (MM-HILIC-ESI-MS^2^) using an anionic pesticide column from Waters and multireaction mode (MRM). In this mode, increased retention time was observed with increasing sugar chain, increasing amount of sialic acids, exchange of N-acetylneuraminic acid (Neu5Ac) by N-glycolyneuraminic acid (Neu5Gc), and shortening of the hydrophobic ceramide anchor (Figure 1B). Using the MRM mode, gangliosides with either Neu5Ac or Neu5Gc were further distinguished by the shift in molecular mass and the mass spectrometric transition to the fragment ions of either [Neu5Ac-H3O]- with *m*/*z* -290 or to [Neu5Gc-H3O]- with *m*/*z* -306 (Figure 1C). For mass spectrometric quantitation, samples were spiked with deuterated gangliosides of known concentration.

### 3.2. Neuroblastoma Have the Highest Expression of GD2 and Also Express Complex Gangliosides but Not N-glycolyl GM3

14 NBL samples (Table 1) were analyzed by TLC and LC-MS^2^. The median age of the patients was 10 months. MYCN amplification, which correlates with an unfavorable outcome [24], was found in two samples. Two samples were diagnosed as ganglioneuroblastoma and ganglioneuroma, respectively. TLC was used to analyze all gangliosides in combination with an anti-GD2 antibody to detect low levels of GD2. In addition, GD2 and N-glycolyl GM3 were quantified by LC-MS^2^. The majority of the samples expressed high levels of GD2 and GT1b (Figure 2A,C). GT1b and its precursor GD1b, which was also found in several samples, are typically present in mature neuronal tissues. Strong GM3 expression was also found in some samples. Most NBL samples showed very high concentration of GD2, which could be detected by chemical orcinol-staining. However, three samples (9, 12, 13) featured low GD2 expression, that was detectable only with an anti-GD2 antibody (Figure 2B). Interestingly, those GD2-low samples were histologically classified as ganglioneuroblastoma (sample 9), differentiating neuroblastoma (sample 12) or ganglioneuroma (sample 13), suggesting a lower percentage of immature neuroblastic tissues. Moreover, the GD2-low samples expressed complex gangliosides of the a-series, particularly GD1a. Quantification of GD2 by LC-MS^2^ (Table 1) correlated quite well with orcinol staining on TLC (R2 = 0.913, Appendix A) and confirmed the heterogeneity in GD2 expression (median 0.54 nmol GD2/mg protein, min 0.04 nmol GD2/mg protein, max 1.93 nmol GD2/mg protein). For one patient, samples were collected at different time points and from different positions (samples 8, 9, and 10). The primary tumor in the adrenal gland (sample 9) was a ganglioneuroblastoma, GD2-low and expressed GD1a. A cerebral metastasis (sample 10) and a lymph node metastasis (sample 8) were poorly differentiated neuroblastoma, had a similar ganglioside profile and expressed GD2. Interestingly, only sample 11 expressed high amounts of both GD2 and GD1a. This sample was histologically defined as a differentiating NBL, but the ganglioside profile suggests the coexistence of GD2-positive immature tissues, and GD1a positive mature tissues. The only two samples classified as 4S that were analyzed in this work (6 and 7) had the highest GT1b expression and rather low GD2 expression, suggesting an evolution toward a mature phenotype. 4S NBLs have indeed a high rate of spontaneous maturation [25]. None of the 14 NBL samples expressed amounts of N-glycolyl GM3 detectable by LC-MS^2^. Indeed, while GM3 with N-acetyl neuraminic acid (GM3(Ac)) was detected, a signal for GM3(Gc) was detected only in mouse liver, used as positive control, but not in NBL samples, independently from the amount of GM3(Ac) (Figure 1D). Taken together, our results indicate that, although GD2 was present in all NBL samples, the level of expression is heterogeneous, with more differentiated samples expressing less GD2.

### 3.3. Medulloblastoma Mainly Express the Gangliosides GD2 or GM3

The ganglioside composition of 9 MB samples was analyzed by TLC and GD2 and N-glycolyl GM3 were quantified by LC-MS^2^ (Table 2). The median age of the patients was 5.5 years. Four MBs were analyzed during routine diagnostics using genome-wide DNA methylation profiling [27] (for MB subtypes and copy number variation profiles we refer to Appendix A). MB expressed very low amounts of gangliosides, in particular of complex gangliosides (Figure 3A). Some MB samples expressed GD2, although to a lesser extent than NBL samples (Table 2). GD2 was visible only by using the immune overlay (Figure 3B). All samples of the SHH subtype (*n* = 4, samples 80, 302, 381 and 436) as well as one group 4 (subgroup V, sample 265) and one group 3 (subgroup II, sample 423) MB were positive for GD2. The median expression was 0.032 nmol GD2/mg protein (min 0.004 nmol GD2/mg protein, max 0.296 nmol GD2/mg protein). The highest expression was found in the sample of group 4. The expression in the SHH group was heterogeneous (from 0.031 to 0.095 nmol GD2/mg protein). The positive sample of group 3 expressed 0.071 nmol GD2/mg protein. Two group 3 (subgroup III and IV) MBs were GD2-negative and expressed GM3 at very high levels. Group 3 and group 4 GD2-positive samples were MYCC and MYCN amplified respectively. However, our cohort is too small to establish a correlation between GD2 and MYC amplification. None of the MB samples expressed detectable amounts of N-glycolyl GM3 by LC-MS^2^. Indeed, only a signal for GM3(Ac) but not for GM3(Gc) was detected, independently from the amount of GM3(Ac) (Figure 1D). CAR-T therapy can result in severe adverse effects due to on-target, off-tumor toxicity, occurring when the target antigen is expressed on both tumor and healthy tissue, particularly if the expression of the antigen in tumor is low [28]. Therefore, because of the low expression of GD2 in MB samples we decided to quantify GD2 also in a relevant brain region. Analysis of three normal pons samples at different ages showed, as expected, that complex gangliosides and sulfatides are strongly expressed. The highest expression of GD2 was detected in the pons of an infant (sample 560) (Figure 3B and Table 3) and was even stronger than in MB samples (0.395 nmol GD2/mg protein in pons vs 0.296 nmol GD2/mg protein in the MB sample with the highest expression). The GD2 expression of the pons samples of the two older donors was within the range of MB samples.

Taken together, our results indicate a subtype-dependent expression of GD2 in MB. While SHH MBs seem to always express GD2, group 3 has a heterogeneous expression. More samples will be required to confirm positive expression in group 4 and low/no expression in the WNT group. GD2 expression in normal pons is within the range or even stronger than in MB.

### 3.4. ST8SIA1 + B4GALNT1 Expression Is a Surrogate Marker for the Detection of GD2-Positive MB in RNAseq Data

While GD2 was always expressed in NBL, some MB samples were GD2- negative (defined as nmol GD2/mg protein < 0.008). To further test our hypothesis of subgroup-dependent GD2 expression in MB, we reanalyzed RNA-Seq data from a recently published MB dataset [18]. First, we performed multiple principal component analyses (PCAs) with varying numbers of genes as part of an unsupervised exploration analysis. Figure 4A shows a PCA calculated on six selected genes, namely *ST3GAL5*, *ST8SIA1*, *ST8SIA5*, *B3GALT4*, *B4GALNT1*, and *B4GALT6* that participate in sphingolipid metabolism of the ganglio series, and which are directly or indirectly involved in the biosynthesis of GD2. The number of genes of the PCA illustrated in Figure 4B was expanded to include the genes of the KEGG Pathway Glycosphingolipid biosynthesis—ganglio series (15 genes). Principal components 1 and 2 of the two PCAs explain over 60% (Figure 4A) and over 50% (Figure 4B) of the variability in the data set, respectively. The five normal tissue samples of the cerebellum (CB) stand out noticeably from the rest of the samples. Due to the different origin of the MB samples, a technical batch effect cannot be excluded. Furthermore, it can be seen in both PCAs that SHH subtype samples also tend to differ from the other groups. Primarily, genes *ST8SIA5*, *ST8SIA1*, and *B4GALNT1* are responsible for the variance along the first principal component. *ST8SIA5* is among others associated with the conversion of GD3 to GT3 and thus a competing enzyme to *B4GALNT1*, which synthesizes GD2 from GD3.

Next, differential gene expression analysis was performed to determine significance between the SHH subtype, which expresses GD2 based on our analysis, and the remaining groups. Figure 5A shows the gene expression of the four genes *B4GALNT1*, *ST3GAL5*, & *ST8SIA1*, and *ST8SIA5*. Especially for *ST8SIA1* and *ST8SIA5*, SHH differed from all other groups. Enzyme *B4GALNT1*, which is involved in the biosynthesis of GD2, showed similarity between SHH and group 4. Recently, a two-gene signature composed of *ST8SIA1* + *B4GALNT1* has been suggested as an efficient predictor of GD2-positive tumors [7]. We applied the two-gene signature on an extended MB dataset composed of the previously used MB dataset and the NBL tumor samples of the TCGA TARGET GTEx study dataset from UCSC Xena. NBL had the highest median values and a large number of the SHH and group 4 subtype samples were in a similar high expression range (Figure 5B). Interestingly, the heterogeneity in the SHH subtype with respect to the two-gene score, correlated with the heterogeneous level of expression as quantified by LC-MS^2^. Single samples of group 3 also showed a high score, similar to what could be seen in TLC and LC-MS^2^. Samples of the WNT subtype had a lower score. Normal tissue samples of the CB had the highest values of the two-gene signature.

Because the two-gene score does not provide a clear threshold for a GD2-positive phenotype, it is necessary to consider the score in the context of different tumor entities and tissues. Similar to the work of the Sorokin et al., group, we placed the MB dataset in the context of three public databases, namely TCGA (Figure 6A), GTEx (Figure 6B), and TARGET (Figure 6C). The clinical data as well as the count values were taken from the TCGA TARGET GTEx study dataset from UCSC Xena study, merged with the MB dataset, and normalized.

The heat map values in Figure 6 are sorted by decreasing order of the two-genes signature. The SHH and group 4 subtypes are listed high up in the TCGA heatmap in their order, lining up alongside brain localized tumor entities such as low grade glioma (LGG) and glioblastoma (GBM), which can be assumed from previous research to be GD2-positive ([17,29]). Group 3 and WNT subtypes are lower in the order (Figure 6A). The normal tissue samples of the CB have the second highest two-gene signature score in the GTEx heatmap and are in the same range of values as other brain areas (Figure 6B). In addition, in Figure 6B, the subtypes group 3 and WNT are further down the scale. In the third heatmap of the TARGET dataset, the two-gene signature score of NBL is as expected higher in order compared to SHH and group 4 (Figure 6C).

Taken together, these data indicate that genes involved in gangliosides synthesis are differentially expressed in MB subgroups and can be exploited to identify GD2-positive samples. Particularly, the application of a two-gene signature suggests that SHH and group 4 are GD2-positive, while expression in group 3 is heterogeneous and WNT is rather GD2 low.

### 3.5. Validation of the Two-Gene Signature Confirms a MB Subtype-Dependent GD2 Expression

RNA-Seq data of the combined expression of *ST8SIA1* + *B4GALNT1* suggested that MB of the subtype SHH and group 4 are positive for GD2. Unfortunately, no TLC data of the same samples were available to verify this prediction. Thus, we aimed to validate the predictive potential of the two-gene signature for the presence of GD2 by analyzing gene expression of *ST8SIA1* and *B4GALNT1* by qRT-PCR in 16 MB samples of different subgroups and in 9 normal brain tissues (Table 3), including exactly the same MB samples as analyzed by TLC and LC-MS^2^. Despite high patient-to-patient variabilities in our cohort, we could confirm that summed expression is higher in GD2+ than in GD2- MB samples (Figure 7A). Interestingly, our data revealed that *ST8SIA1* + *B4GALNT1* expression significantly differs in pediatric compared to adult normal brain tissue (Figure 7B), which might reflect a change in GD2 expression during brain maturation as also shown by TLC of pons (Figure 3, Table 3). We then plotted the single expression levels of both genes in a stacked graph to be able to correlate the mRNA levels with the GD2 status of the individual samples. We also included samples without a known value of GD2 expression (Figure 7C). Tumors of the SHH group as well as of group 4 showed the highest combined expression of *ST8SIA1* and *B4GALNT1*. This suggests that the high levels of *ST8SIA1* + *B4GALNT1* in both groups may be indeed indicative of the availability of GD2. In MB of group 3 and WNT the two-gene signature seems to not strictly correlate with GD2-positive or -negative status as samples 25 and 423 had a similar value but were GD2-negative and positive, respectively. *ST8SIA1* + *B4GALNT1* levels of MB samples that could not be clearly assigned to group 3 or 4 and are therefore referred to as “group 3/4” were low compared to group 4 and thus resembled the expression of group 3 MB. The value of the two genes combination, however, did not correlate to the amount of GD2 in this small number of samples (Figure 7D).

Taken together, these results confirm that particular MB of the SHH and the group 4 subtype have a high combined expression of *ST8SIA1* + *B4GALNT1*. Group 3 and WNT have a rather low combined expression.

## 4. Discussion

Molecular analysis of pediatric tumor samples has been shown to increase the survival of children with aggressive tumors. However, in about 57% of pediatric patients a molecular target cannot be identified, even after the application of sophisticated methods such as whole-genome sequencing (WGS) and transcriptomics [11]. Targeted therapies for medulloblastoma are limited, due in part to the lack of targetable somatic single-nucleotide variants and the low mutational burden. Preliminary preclinical and clinical results suggest potential therapeutic benefits for the treatment of brain malignancies with CAR-T cells therapy [30], and clinical studies with CAR-T cells targeting GD2 are enrolling also MB patients. Our data suggest that particularly MB patients of the SHH and group 4 subtype could benefit from GD2-directed therapies and that the ganglioside profile could help in refining the MB classification. However, our data also suggest that expression of GD2 in the normal brain should be considered for risk stratification prior to GD2-CAR-T cell therapy.

### 4.1. TLC and LC-MS^2^ Based Analysis of Gangliosides

GD2 and N-glycolyl GM3 expression is described in several pediatric and adult tumor entities, whereby different methods have been applied [31,32,33,34]. However, the accurate determination and, thus, the applicability for diagnostic and therapeutic purposes is challenging. Several studies used GD2 and N-glycolyl GM3 staining on FFPE tissues, even if the method is controversial because gangliosides are soluble in solvent such as ethanol that is required for the preparation of FFPE samples [35]. Moreover, cross reactivity of an antibody with other ganglioside structures and similar epitopes on glycoproteins, which may give a false positive result, need to be ruled out carefully. GD2 expression has also been analyzed by flow cytometry on cultivated cells [36], but GD2 levels may be influenced by the culture conditions [37]. More sophisticated methods such as radioimmunoscintigraphy with an ^131^I-radiolabeled anti-GD2 monoclonal antibody (mAB), [38] allow the direct detection of GD2-positive tumors in patients, but this method is available only in a few institutions. In our study we performed TLC of extracts from frozen samples together with quantification by LC-MS^2^. TLC represents a cheap and reliable methodology, particularly combined with immune overlay, to assess GD2 expression on native samples. The method also allows detection of other gangliosides. This is important to understand changes in ganglioside profiles under anti-GD2 therapy and to identify subtype-specific profiles. Importantly, we found a good correlation between the amount of GD2 quantified by LC-MS^2^ and orcinol staining. LC-MS^2^ is a more sophisticated and expensive methodology which allows the exact quantification of gangliosides and particularly the identification of specific modifications, such as N-glycolyl sialic acid-containing GM3. Moreover, LC-MS^2^ also allows the analysis of the composition and length of the ceramide anchor, which is known to change between differentiation stages and cell types [39,40,41]. However, quantification requires the use of internal stable isotope-labelled ganglioside standards. Their commercial availability is limited so far and the analysis is possible only in specialized laboratories. According to our LC-MS^2^ assay, NBL and MB do not express detectable amounts of N-glycolyl GM3. In a previous study N-glycolyl GM3 expression was observed in 81% of NBL samples by using immunohistochemistry with a NeuGc-GM3-specific antibody (14F7 murine monoclonal antibody) on FFPE tissues [34]. With our protocol. N-glycolyl and N-acetyl neuraminic acid carrying gangliosides are detected with the same mass spectrometric transition, i.e., cleavage of the same bond. They are simply distinguished by mass differences. Hence, a bias towards one or the other species in our protocol can be excluded. This is reflected by the intense signals, which we obtained for N-glycolyl GM3 and N-glycolyl GM2 in mouse liver. However, the antibody-based assay may detect significantly lower amounts of this ganglioside. Further analysis is required to correlate the quantity of tumor GD2 and/or N-glycolyl neuraminic acid carrying gangliosides with a clinical response to GD2 and/or N-glycolyl GM3-directed therapies.

### 4.2. Gangliosides Patterns in NBL and MB

Synthesis of the haemato-series gangliosides GM3 and GD3 predominates during early embryogenesis of vertebrates, whereas the synthesis of the more complex gangliosides, such as GM1, GD1a, GD1b, and GT1b, predominates at later embryogenic stages. This implies that gangliosides can be useful stage-specific marker molecules in developing cells, including embryogenesis and stem cells [42]. NBL cells can differentiate into mature neurons and different stages of differentiation may be present in the same tumor [43]. Accordingly, we detected a very complex ganglioside profile in neuroblastoma with a mixture of early embryogenesis gangliosides, such as GD2, and gangliosides that are normally present in mature neuronal tissues, such as GD1b and GT1b. Among our specimen, samples with a low concentration of GD2 were classified as ganglioneuroma and ganglioneuroblastoma that are characterized by a higher number of differentiated cells. Interestingly, the a-series (GM1 and the complex ganglioside GD1a) was expressed prevalently in samples with low GD2 expression which co-expressed complex gangliosides of a- and b-series. Mature neurons express the a- and b-series of complex gangliosides but very low amounts of GD2 [44]. Thus, samples with low GD2 expression and co-expression of gangliosides belonging to the a- and b-series may represent a more mature phenotype. Rather, undifferentiated tumors would lack complex gangliosides, and Schengrund et al., reported loss of complex GT1b to correlate with poor prognosis for neuroblastoma patients [45].

In contrast to NBL, MB preferentially expresses simple gangliosides, suggesting that they belong to a more primitive and undifferentiated stage. Interestingly, within group 3, some samples were strongly GM3 positive but GD2-negative. Considerable heterogeneity exists within the four MB subgroups, reflecting differences in prognosis and outcome. Integrated multi-omics data including DNA methylation profiles have identified subtypes within each subgroup of MB [46,47]. Lipid analysis has not been integrated so far to define MB subtypes, although gangliosides have been discussed in the past as potential markers for classification and grading of CNS tumors [29]. Our results suggest that GM3 and GD2 could be helpful to further characterize MB subtypes particularly within the group 3 and based on the RNA-Seq results of the two-gene signature also within the SHH group. Changes in the ganglioside composition might be correlated to the differentiation stage within the subtype. Indeed, group 3 subtypes can be distinguished based on the relative proportion of immature primitive progenitor-like or more mature neuronal-like cells [48].

### 4.3. Relevance of Gangliosides Expression for Therapy of MB and NBL

In this study, MB expressed less GD2 than NBL. However, the amount of GD2 as quantified by LC-MS^2^ was very heterogeneous within MB and NBL samples. So far, there are no preclinical or clinical data defining the level of GD2 expression required to trigger antitumor responses when using monoclonal antibodies. However, low percentage of GD2-positive cells before immunotherapy was associated with relapse in NBL patients receiving anti-GD2 antibodies [49]. Disappointing results have been achieved with anti-GD2 antibodies in tumor entities other than NBL for example in SCLC [50] and metastatic melanoma [51]. Importantly, ganglioside composition in melanoma samples is very heterogeneous, and GD2 is far less expressed compared to GM3 and GD3 [37]. According to the same study, which also analyzed the ganglioside composition by TLC, in 60% of the melanoma samples GD2 was not detectable by orcin staining, and only in 2% of the samples a strong GD2 expression was present. Other immunotherapies could be used for targeting cancer cells with low expression of GD2. CAR-T cells can recognize lower antigen densities compared to monoclonal antibodies [52] and it is to be expected that CAR-T cells against GD2 will be more successful for treating tumors with low concentration of GD2. Moreover, CAR-T cells are able to cross the blood-brain barrier and therefore should be particularly effective in the treatment of brain tumors. However, our data suggest that not all MB patients will benefit from a GD2-directed therapy. While the WNT subtype has a good prognosis, the other three groups have a mixed response [53]. Particularly, group 3 has the highest relapse rates and after standard-of-care have a 20% survival rate. While SHH and probably group 4 MB patients are eligible for CAR-T- GD2 clinical studies, GD2 analysis should always be performed before inclusion of group 3 MB. In case of GD2 negativity, these patients may benefit from other CAR-T cells studies. Several antigens are currently under investigation for immunotherapy of MB including B7-H3 EPHA2, HER2 and interleukin 13 receptor α2 [30,54]. Importantly, our data shows expression of GD2 in normal pons and GD2 expression in other brain regions such as cerebellum has been described in the literature [55]. Toxicity related to CAR-T-GD2 treatment due to the recognition of normal brain regions by the modified T cells is a matter of discussion in the literature with conflicting results [56,57]. Our results indicate that the age of patients should be considered when assessing the risk associated with the application of CAR-T cells.

GD2-negative MB samples were strongly positive for GM3. GM3 has a wide expression also in normal tissues and immunotherapy has concentrated mostly on targeting tumor-specific modifications, particularly N-glycolyl GM3 via racotumomab. After a successful phase II/III study, racotumomab was conditionally approved in Latin American countries as maintenance therapy for advanced NSCLC [58]. In our study, only two samples were strongly GM3 positive, but N-glycolyl-containing sialic acid was not detected. More samples will be necessary to confirm the absence of NeuGc-GM3 in MB.

In postnatal brain, synthesis of simple gangliosides is switched to the synthesis of complex, brain-type gangliosides resulting in terminal differentiation and loss of “stemness” of neuronal stem cells. Consequently, our research suggests that the modulation of stage-specific gangliosides could also represent a therapeutic approach to influence fate determination and finally cell proliferation [36,59].

Finally, the influence of the ganglioside pattern on anti-GD2 directed therapies should be further investigated. Gangliosides do indeed have immunosuppressive activities. GM3, which according to our data is particularly abundant in MB group 3, inhibits Natural Killer (NK) cells cytotoxicity [60]. NBL-derived gangliosides inhibit the function of T cells and dendritic cells supporting the escape of tumor cells from immune recognition and elimination [61]. Thus, the ganglioside composition rather than the expression of GD2 alone should be studied to understand patient-specific answers to immunotherapy.

### 4.4. Biomarkers for the Detection of GD2-Positive MB

The only recognized surrogate markers for the identification of GD2-positive tumors to date are the H3F3A K27M and the HIST1H3B K27M mutations in diffuse midline gliomas (DMG) [62]. However, such mutations are found almost exclusively in DMG and cannot be used as surrogate markers in other tumor entities. The expression of genes required for GD2 synthesis, particularly *B4GALNT1*, has been discussed in the literature as a surrogate marker for the identification of GD2-positive NBL [63]. Because of the complexity of the pathway required for ganglioside synthesis, several algorithms have been discussed in the literature to predict GD2 expression. The group of Sorokin proposed that transcriptome data can be used to predict GD2 expression based on the expression of *ST8SIA1* and *B4GALNT1* [7]. Based on this signature, they predicted high GD2 expression in brain tumors. However, MB samples were not analyzed. Our analysis suggest that the algorithm may help in identifying GD2-positive MB. However, even if the score can identify GD2-positive normal tissues and tumors, the amount of GD2 is difficult to predict. Moreover, further downstream metabolism of GD2 in the b-series, differential expression of the individual ganglio-series, and glycolipid turnover rates, as well as post-transcriptional regulatory mechanisms are not taken into account by the proposed two-gene signature. The inclusion of further genes will be probably required to increase the reliability of the score. Recently, an alternative approach was followed [64] in which the glycosyltransferases *ST3GAL2*, *ST3GAL3*, *B4GALT5*, and *B3GALT4* were suggested as a further predictor in identifying GD2-positive phenotypes in cancer patients.

### 4.5. Limitations of the Study

The number of the samples included in the study (14 NBL and 9 MB samples) and particularly the number of MB available per subgroup are clearly too low for a generalization of all the results. However, we can confirm that NBL are GD2 positive, even if the heterogeneity of the expression within samples is very high, while some MB are GD2 negative. The GD2 expression in NBL is higher compared to MB. Moreover, we identified ganglioside profiles such as the GM3-positive/GD2-negative phenotype in group 3 MB and the GD1a-positive/GD2-low phenotype in NBL which should be further characterized in a larger cohort. We cannot exclude that the GM3-positive/GD2-negative phenotype also exists in other MB subtypes because only one sample was analyzed by LC-MS^2^ and TLC in WNT and group 4. Similarly, we cannot conclude that group 4 and WNT MB are generally GD2 positive and negative respectively, even if suggested by the two-genes score. Due to the difficulty to access autopsy material of good quality from children, GD2 quantification was performed only in pons and more brain regions from donors of different ages should be analyzed in further studies. Moreover, TLC and LC-MS^2^ cannot distinguish whether GD2 is generally low expressed in all tumor cells or only expressed in some cells. Immunohistochemistry with anti-GD2 antibody on frozen sections or the upcoming technology of single cell lipidomics will help in the future to identify which cell subtypes express GD2. Finally, more complex algorithms will be required to more accurately predict the quantity of GD2 starting from RNAseq data.

## 5. Conclusions

Precision medicine is driven by the idea of integrating clinical data with patient specific multi-omics data to develop therapeutic strategies. So far, genomic, transcriptomic, and methylation data have been used to identify therapeutic targets and to define molecular subtypes with prognostic and therapeutic potential. The implementation of ganglioside profiling and quantification has the potential to identify patients who may benefit from therapy against lipid targets. Surrogate biomarkers predicting their expression may facilitate the identification of suitable patients in the future. The utility of ganglioside profiles for the classification and grading of tumors, particularly MB, should be further analyzed.

## Figures and Tables

**Figure 1 cancers-14-06051-f001:**
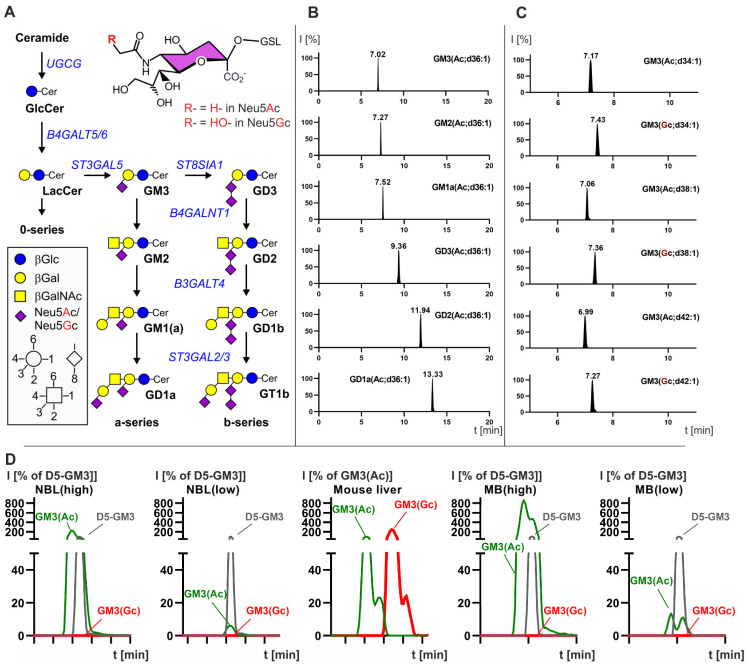
Biosynthesis and liquid chromatography-coupled tandem mass spectrometric detection of gangliosides. (**A**) Brain gangliosides are derived via glucosylation of ceramide. GM3 is substrate for two enzymes encoded by the genes *B4GALNT1* and *ST8SIA1*, which either elongate the glycan chain by an N-acetylgalactosamine residue or a sialic acid moiety. This guides ganglioside biosynthesis either into a- or b-series complex gangliosides. Most commonly, sialic acid is N-acetyl neuraminic acid (Neu5Ac). However, some tumors also integrate N-glycolyl neuraminic acid (Neu5Gc), likely derived through nutritional uptake. (**B**,**C**) Detection of gangliosides with mixed mode hydrophilic interaction chromatography-coupled tandem mass spectrometry. I, Intensity of the mass spectrometric signal. (**B**) With increasing glycan moiety gangliosides elute later from the column. Furthermore, they are identified in negative mode MRM by their molecular ion size (*m*/*z*) and their collisional transition to a specific sialic acid fragment. (**C**) Gangliosides GM3 containing Neu5Ac are detected in MRM mode with a transition to the [Neu5Ac-H30]- ion (*m*/*z* 290), whereas those containing Neu5Gc are recorded setting the transition to the [Neu5Gc-H3O]- ion with *m*/*z* 306. Note that GM3 species with Neu5Gc elute later than corresponding species with Neu5Ac, but the increasing ceramide anchor size shifts GM3 to slightly earlier retention times. (**D**) Exemplified overlay of LC-MS^2^ MRM detections for GM3 with N-acetyl neuraminic acid (GM3(Ac)) and with N-glycolyl neuraminic acid (GM3(Gc)) for medulloblastoma and neuroblastoma samples with relatively low or high expression of GM3. Mouse liver sample served as positive control for the detection of GM3(Gc).

**Figure 2 cancers-14-06051-f002:**
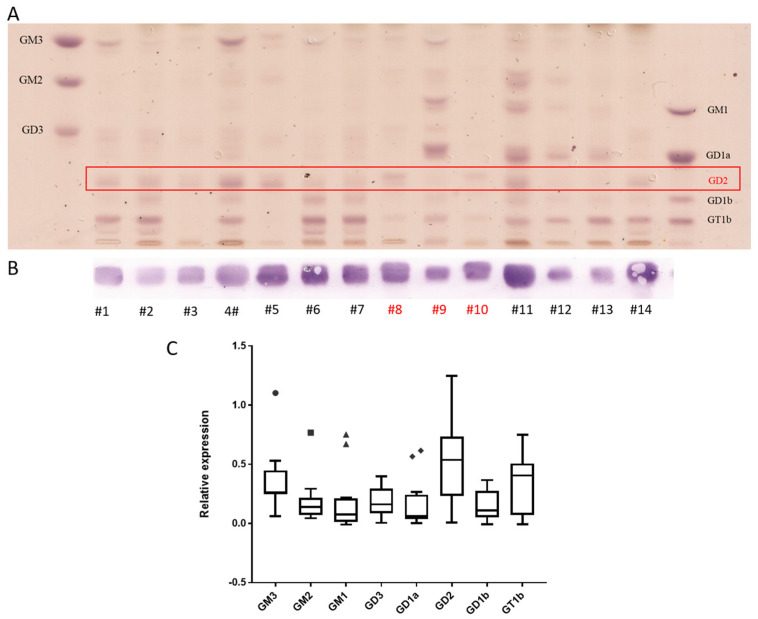
NBL expresses high amounts of GD2 and complex gangliosides. The acidic fraction of gangliosides was separated and stained with orcinol (**A**) or incubated with an anti-GD2 antibody (**B**). Purified gangliosides were used as standards. (**C**) Relative quantification of gangliosides based on TLC shown as box and whiskers plots (in the style of Tukey). Samples #8, #9, and #10 (in red) belong to the same patient. Boxes range from the first to third quantiles, divided by a line indicating the median, with whiskers demonstrating the largest.

**Figure 3 cancers-14-06051-f003:**
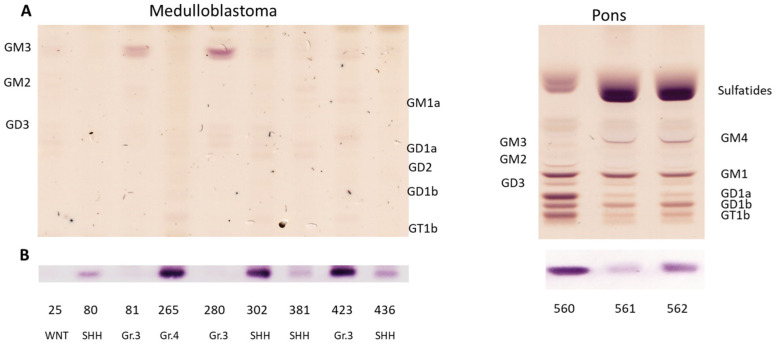
GD2 is expressed at a low level in some MBs and normal pons. The acidic fraction of gangliosides was separated and stained with orcinol (**A**) or incubated with an anti-GD2 antibody (**B**). The subgroup of the MB samples is indicated.

**Figure 4 cancers-14-06051-f004:**
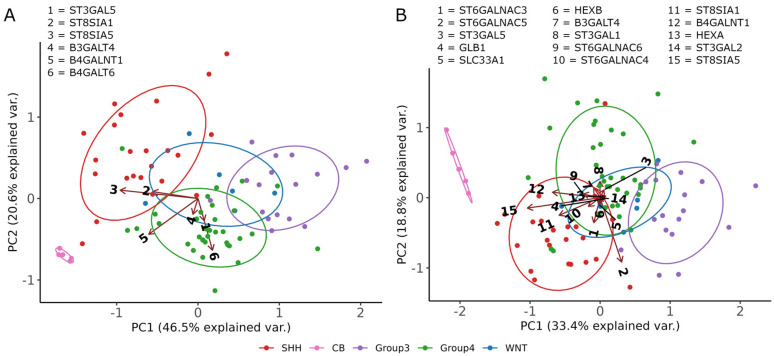
Principal component analysis of gene expression data of six selected genes (**A**) and genes of the Glycosphingolipid biosynthesis—ganglio series pathway retrieved from KEGG (**B**). Individual samples are represented by colored dots. The color of a dot defines the MB subtype (SHH, Group 3, Group 4, and WNT) or normal cerebellum (CB) of a sample. The ellipses represent the core area of the subtypes by the confidence interval of 68%. The arrows are projections of the original basis vectors (the variables) onto the PC plane. High resolution versions of the PCA plots are in Appendix A.

**Figure 5 cancers-14-06051-f005:**
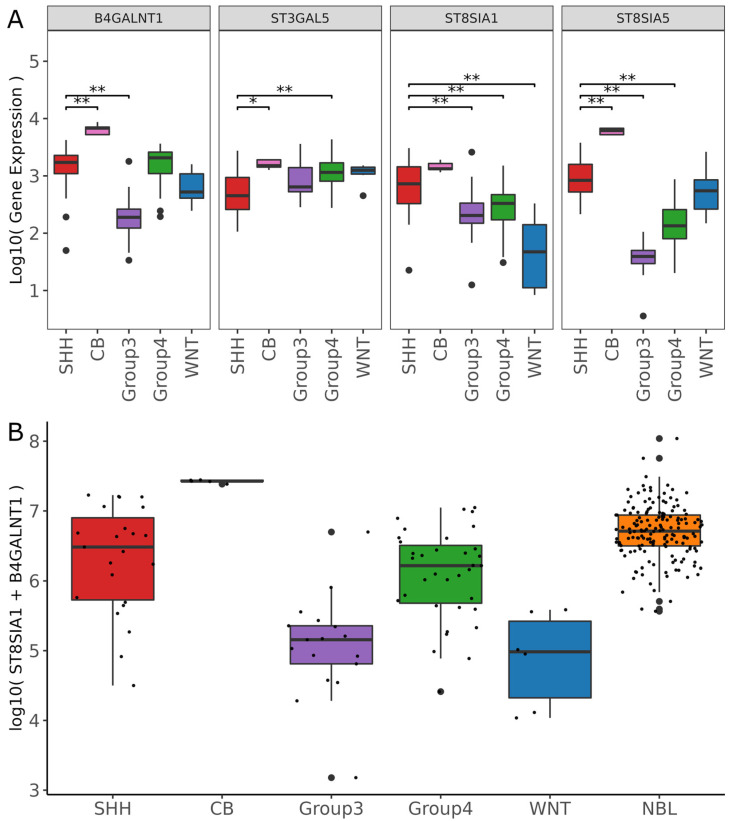
(**A**) Boxplots showing gene expression values of individual genes (*B4GALNT1*, *ST3GAL5*, ST*8SIA1*, and *ST8SIA5*) grouped by MB subtypes (SHH, Group 3, Group 4, WNT) and normal cerebellum (CB). ** = *p* < 0.01. * = *p* < 0.5 represent the *p*-adjusted values derived from the DGE analysis by DESes2 package. The Wald test was used for significance testing. (**B**) Boxplots showing the two-gene score (*B4GALNT1* + *ST8SIA1*) grouped by MB subtypes and NBL sample type derived from the TCGA TARGET GTEx study dataset from UCSC Xena. Boxes range from the first to third quantiles, divided by a line indicating the median, with whiskers demonstrating the largest and lowest values no further than 1.5 * IQR from the hinge. In (**B**) the dots represent individual samples. Gene expression values are log10-transformed and normalized by the median of ratios method.

**Figure 6 cancers-14-06051-f006:**
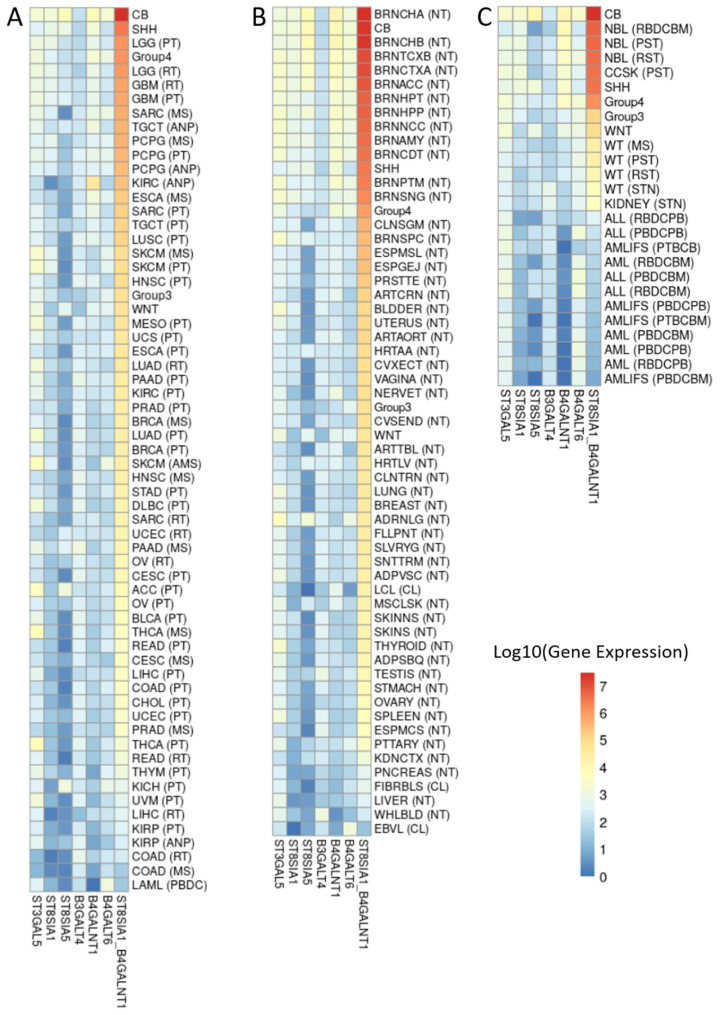
Heatmaps showing the gene expression of six enzymes (*ST3GAL5*, *ST8SIA1*, ST8SIA5, *B3GALT4*, *B4GALNT1*, and *B4GALT6*) involved in the ganglioside biosynthesis, and the two-gene signature score composed of the sum of *ST8SIA1* & *B4GALNT1* genes. Color scale indicates log10-transformed and normalized by the median of ratios method gene expression values. Sample types of each heatmap are sorted in descendent order by the two-gene signature score. Sample metadata and RSEM expected count data are obtained from the TCGA TARGET GTEx study dataset from UCSC Xena. The dataset was divided in the following subsets: (**A**) TCGA project, (**B**) GTEx project, (**C**) TARGET project, and merged with the MB dataset of 86 samples. A detailed transcription of the sample type abbreviations is provided in Appendix A.

**Figure 7 cancers-14-06051-f007:**
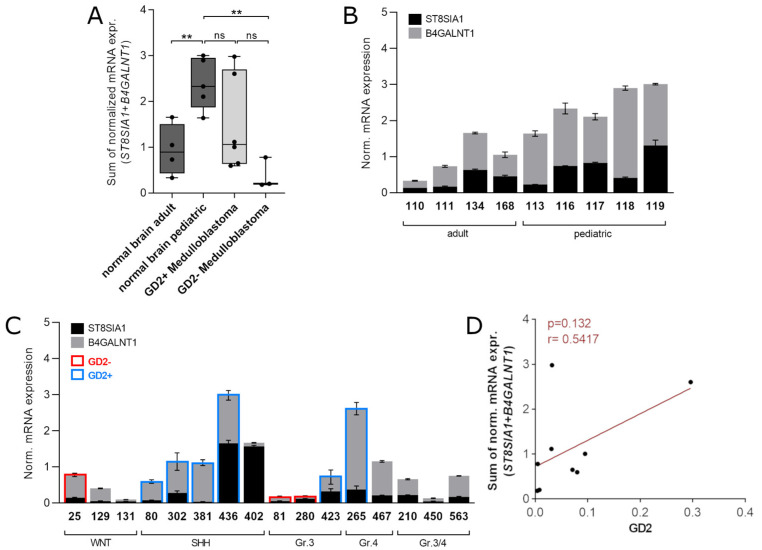
*ST8SIA1* + *B4GALNT1* combined expression is high in GD2-positive samples and is dependent on the MB subtype. (**A**,**B**) *ST8SIA1* + *B4GALNT1* combined expression in normal brain samples of adult and pediatric patients and in MB in GD2+ and GD2- MB samples. Combined expression was calculated as sum of normalized expression values of single genes. (**C**) *ST8SIA1* and *B4GALNT1* expression of individual patient samples of normal brain and MB. GD2 status according to LC-MS^2^ analysis is color-coded where available. ** = *p* < 0.01. In B and C the mean and standard deviation of technical replicates is indicated. (**D**) Correlation between GD2 expression (in nmol/mg protein as measured by LC-MS^2^) and the combined expression of *ST8SIA1* and *B4GALNT1* as measured by qRT-PCR.

**Table 1 cancers-14-06051-t001:** Neuroblastoma patients and sample characteristics.

Nr	Age	Sex	Histology	Localization	Sample Type	INSS	MYCN	GD2
#1	5 m	m	NBL, pd	adrenal gland	PT	nd	NA	0.64
#2	1 m	f	NBL, pd	adrenal gland	PT	1	NA	0.95
#3	1 y	m	NBL, pd	retroperitoneal	PT	3/4	A	0.59
#4	5 m	m	NBL, pd	adrenal gland	PT	nd	nd	1.93
#5	1 y	m	NBL, pd	retroperitoneal	PT	3/4	A	1.06
#6	8 m	m	NBL, pd	lumbal soft tissue	PM	4S	NA	0.49
#7	2 m	m	NBL, pd	adrenal gland	PT	4S	NA	0.43
#8	8 y	m	NBL, pd	lymphnode	PM	4	NA	1.1
#9	8 y	m	GNB, int	adrenal gland	PT	4	NA	0.16
#10	10 y	m	NBL, pd	intracerebral	R	4	nd	0.31
#11	8 y	m	NBL, dif	retroperitoneal	PT	4	nd	1.58
#12	2 y	f	NBL, dif	para/intraspinal	PT	nd	NA	0.15
#13	4 y	f	NBL, dif/mature GN	adrenal gland	PT	nd	nd	0.04
#14	5 m	f	NBL, pd	adrenal gland	PT	nd	NA	0.49

m/y: months/years; f: female; m: male; NBL: Neuroblastoma; pd: poorly differentiated; int: intermixed; dif: differentiating; GNB: Ganglioneuroblastoma; GN: Ganglioneuroma; PT: primary tumor (at initial diagnosis); PM: primary metastasis (at initial diagnosis); R: relapse; INSS: International Neuroblastoma Staging System [26]; nd: not determined; NA: not amplified; A: amplified. Sample 8, 9 and 10 belong to the same patient. The amount of GD2 is in nmol/mg protein as measured by LC-MS^2^.

**Table 2 cancers-14-06051-t002:** Medulloblastoma patients and sample characteristics.

Nr	Age	Sex	Histological Subtype	Molecular Subtype	Sample Type	MYCC	MYCN	GD2
25	15 y	f	CBM	WNT	PT	nd	nd	0.005
80	6 y	f	AMB	SHH	PT	nd	nd	0.080
81	5 y	m	CBM	Gr.3/III	PT	nd	NA	0.004
129	8 y	f	CBM	WNT	PT	nd	nd	nd
131	8 y	f	CBM	WNT	PT	nd	nd	nd
210	5 y	m	CBM	Gr.3/4	PT	NA	NA	nd
265	11 y	m	AMB	Gr.4/V	PT	NA	Gain	0.296
280	6 y	f	CBM	Gr.3/IV	PT	NA	NA	0.008
302	1 y	f	DMB	SHH	PT	NA	NA	0.095
381	1 y	f	DMB	SHH	PT	nd	nd	0.031
402	1 y	f	DMB	SHH	R	nd	nd	nd
423	6 y	m	CBM	Gr.3/II	PT	A	NA	0.071
436	3 y	f	DMB	SHH	PT	NA	NA	0.032
450	2 y	f	CBM	Gr.3/4	PT	NA	NA	nd
467	15 y	f	CBM	Gr.4	PT	NA	NA	nd
563	5 y	m	CBM	Gr.3/4	PT	NA	NA	nd

y: years; f: female; m: male; CMB: classic medulloblastoma; AMB: anaplastic medulloblastoma; DMB: desmoplastic medulloblastoma; PT: primary tumor (at initial diagnosis); R: relapse; nd: not determined; NA: not amplified; A: amplified. Samples that could not be clearly assigned to group 3 or 4 were designated as “group 3/4”. All samples were localized in Posterior Fossa. Samples #129 and #131 as well as #381 and #402 is material from the same patient, respectively, obtained at different time points. The amount of GD2 is in nmol/mg protein as measured by LC-MS^2^.

**Table 3 cancers-14-06051-t003:** Patient information & characteristics of normal brain samples.

Sample	Age	Sex	Localization	GD2
110	22, 26, 27, 28, 29 y	m	Frontal lobe	nd
111	Adult (age unknown)	unknown	Temporal lobe	nd
113	2 y	m	Temporal lobe	nd
116	2 y	m	Hippocampus	nd
117	8 m	m	Hippocampus	nd
118	8 m	m	Temporal lobe	nd
119	2 y	m	Brain stem	nd
134	24 y	m	Cerebellum	nd
168	60 y	f	Pons	nd
560	1 d	m	Pons	0.395
561	29 y	m	Pons	0.065
562	61 y	m	Pons	0.159

d/m/y: days/months/years; f: female; m: male; nd: not determined. Samples 113, 116 and 119 as well as 117 and 118 is material from the same patient, respectively, obtained from different brain regions. The amount of GD2 is in nmol/mg protein as measured by LC-MS^2^.

## Data Availability

Data supporting the reported results can be obtained from the corresponding author.

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
