# Peer review of "GD2 Expression in Medulloblastoma and Neuroblastoma for Personalized Immunotherapy: A Matter of Subtype"

_cancers, 2022, doi:10.3390/cancers14246051_

Round 1

Reviewer 1 Report

The role of gangliosides in tumor biology and as a treatment target has seen little attention largely due to the difficulty in measuring these molecules. Here, the authors go some way to addressing this by performing a thorough analysis of ganglioside expression in neuroblastoma and medulloblastoma. GD2 targeted therapy in neuroblastoma is standard of care, and the inclusion of this tumor type in the present study acts as a valuable reference against which ganglioside expression levels in medulloblastoma and other tumors can be assessed for therapeutic potential.

Methods and data presented in the manuscript are good and provide a useful resource for other researchers in the field. I only have minor suggestions:

Figure 2 – perhaps show a correlation of GD2 orcinol staining versus anti-GD2 antibody staining and report R value and significance.

Figure 4 – need to modify the figures so the gene name are readable for all.

Author Response

We thank the reviewer for the positive feedback

Figure 2 – perhaps show a correlation of GD2 orcinol staining versus anti-GD2 antibody staining and report R value and significance.

We performed a correlation of GD2 orcinol staining versus anti-GD2 antibody staining (Fig. S1A) and in addition of GD2 orcinol staining versus LC-MS2 quantification (Fig. S1B) and report corresponding R squared values and 95% confidence bands in supplemental Figure S1. Whereas the orcinol staining on TLC correlates quite well with the quantitative and specific LC-MS2 quantification of GD2, the anti-GD2 detection, which was set up as a rather qualitative assay, does not correlate well. Technical difficulties with the anti-GD2 immune overlay assay are in part reflected by rather equally thick bands, which are sometimes stained rather faint and sometimes rather intense although their area size would imply more or less similar amounts (e.g. Figure 2B sample #4 and #5). We commented the result at line 317 in the manuscript (PDF Version, line 323 doc Version). Moreover, by reanalysing the LC-MS2 data for calculating the correlation to orcinol, we realised that one value was reported wrongly in Table 1 and we corrected it.

Figure 4 – need to modify the figures so the gene name are readable for all.

Thank you for the suggestion. We changed the Figure as suggested. Additionally, we provide high resolution versions of the PCA plots in Fig S3

Reviewer 2 Report

The authors studied the presence of gangliosides in neuroblastoma and medulloblastoma. They report the ganglioside subtypes between these tumors and their corresponding tumor subtypes. The authors use readily available techniques including TLC to determine the expression of GD and subtypes in these various tumor types. This avoids the issue of degraded GD from FFPE sections. 

MAJOR: 

1. The discussion is too long and should be condensed. The authors should include a paragraph or at least a few sentences describing potential weaknesses of the study including generalizability of their results. 

MINOR: In the simple summary and abstract and discussion, the use of the word "profit" is an odd choice. Would recommend the word "benefit".  

Page 3, line 102, this sentence is awkward:  "Indeed, in a recent study with 23 MBs no very 102 high evidence molecular targets for personalized therapy were identified."  Would recommend rewording this sentence. 

Author Response

 We thank the reviewer for the positive feedback

MAJOR: 

The discussion is too long and should be condensed. The authors should include a paragraph or at least a few sentences describing potential weaknesses of the study including generalizability of their results. 

We have condensed the discussion, particularly we included the paragraph on N-glycolyl GM3 in paragraph 4.1 and included a paragraph on the limitations of our study

MINOR: In the simple summary and abstract and discussion, the use of the word "profit" is an odd choice. Would recommend the word "benefit".  

Thank you for this comment. We have changed profit to benefit throughout the manuscript

Page 3, line 102, this sentence is awkward:  "Indeed, in a recent study with 23 MBs no very 102 high evidence molecular targets for personalized therapy were identified."  Would recommend rewording this sentence. 

We changed the sentence to: “Indeed, in a recent study with 23 MBs no clinically relevant molecular targets were identified”

Reviewer 3 Report

The authors have no control. Medulloblastoma and neuroblastoma are cancers that are located differently. For this reason, they should be compared with different controls. Medulloblastoma should be compared with cerebella samples or adjacent normal nerve tissue. The authors only have controls where there is only one cerebella sample of a person who was 24 years old. Medulloblastoma is a childhood cancer and should not be compared to an adult sample. Control samples are mainly CNS samples in the vast majority of adults. Neuroblastoma is not located in the CNS and is a childhood cancer.

Some analyzed subgroups count only one sample. This is not enough to draw any conclusions in a scientific article.

Abbreviations entered incorrectly. There is an unexplained abbreviation SHH in the abstract.

The abstract is 235 words long. According to the guidelines, it should be a maximum of 200 words.

The methodology lacks a subchapter on statistical analysis. The results shown in the article should be with measurement uncertainty.

The results of the amount of GD2 in the tested tumors should be compared with the level of GD2 from the controls.

The authors in Figure 2 analyze the level of various gangliosides. The results should be compared with the control.

Author Response

The authors have no control. Medulloblastoma and neuroblastoma are cancers that are located differently. For this reason, they should be compared with different controls. Medulloblastoma should be compared with cerebella samples or adjacent normal nerve tissue. The authors only have controls where there is only one cerebella sample of a person who was 24 years old. Medulloblastoma is a childhood cancer and should not be compared to an adult sample. Control samples are mainly CNS samples in the vast majority of adults. Neuroblastoma is not located in the CNS and is a childhood cancer.

The aim of this work was not to compare neuroblastoma and medulloblastoma to the normal counterpart but to use neuroblastoma as a reference against which ganglioside expression levels in medulloblastoma can be assessed for therapeutic potential. Indeed, GD2 targeted therapy in neuroblastoma is standard of care while it was not successful so far in other tumour entities. GD2 quantification in normal pons was also performed because on-target, off-tumor toxicity of CAR-T cells against GD2 has been discussed in the literature due to expression of GD2 in normal brain. We agree that it would be of interest to analyse the gangliosides expression also in cerebellum and further brain regions of children, however such samples are difficult to obtain because autopsies in children in Germany are performed rarely and moreover the tissues are often degraded. We have therefore discussed results on GD2 expression in cerebellum based on the literature (line 648, PDF version, line 692 doc version)

Some analyzed subgroups count only one sample. This is not enough to draw any conclusions in a scientific article.

We have commented this in the new paragraph: limitations of the study, from line 694, PDF file (line 740, doc file)

Abbreviations entered incorrectly. There is an unexplained abbreviation SHH in the abstract.

We have changed it in the abstract

The abstract is 235 words long. According to the guidelines, it should be a maximum of 200 words.

We shortened the abstract.

The methodology lacks a subchapter on statistical analysis. The results shown in the article should be with measurement uncertainty.

The statistical analysis is always described together with the related methodology. We think that this way, it is easier for the readers. Additionally, we have included SD values in Figure 7.

The results of the amount of GD2 in the tested tumors should be compared with the level of GD2 from the controls. The authors in Figure 2 analyze the level of various gangliosides. The results should be compared with the control.

As stated before, the aim of this work was not to compare neuroblastoma and medulloblastoma to the normal counterpart but to use neuroblastoma as a reference against which ganglioside expression levels in medulloblastoma can be assessed for therapeutic potential.

Round 2

Reviewer 3 Report

The authors draw conclusions from the study of groups of one case (Gr.3/III, Gr.4/V, Gr.3/IV). This is unacceptable in a scientific article. The authors should collect a larger group of patients, analyze the level of GD2 and then count with a statistical test whether the groups differ.

Authors must use an appropriate control group. If they do not have a sample from non-cancer patients, they should use adjacent nerve tissue as a control.